# Secondary Emission Calorimetry

Burak Bilki [1,2,3,*], Kamuran Dilsiz [4], Hasan Ogul [5], Yasar Onel [2], David Southwick [2], Emrah Tiras [6], James Wetzel [2] and David Roberts Winn [7]

1 Department of Mathematics, Beykent University, Istanbul 34500, Turkey
2 Department of Physics and Astronomy, University of Iowa, Iowa City, IA 52242, USA
3 Turkish Accelerator and Radiation Laboratory, Ankara 06830, Turkey
4 Department of Physics, Bingöl University, Bingöl 12000, Turkey
5 Department of Nuclear Engineering, Sinop University, Sinop 57000, Turkey
6 Department of Physics, Erciyes University, Kayseri 38030, Turkey
7 Department of Physics, Fairfield University, Fairfield, CT 06824, USA
* Correspondence: burak.bilki@cern.ch

**Abstract:** Electromagnetic calorimetry in high-radiation environments, e.g., forward regions of lepton and hadron collider detectors, is quite challenging. Although total absorption crystal calorimeters have superior performance as electromagnetic calorimeters, the availability and the cost of the radiation-hard crystals are the limiting factors as radiation-tolerant implementations. Sampling calorimeters utilizing silicon sensors as the active media are also favorable in terms of performance but are challenged by high-radiation environments. In order to provide a solution for such implementations, we developed a radiation-hard, fast and cost-effective technique, secondary emission calorimetry, and tested prototype secondary emission sensors in test beams. In a secondary emission detector module, secondary emission electrons are generated from a cathode when charged hadron or electromagnetic shower particles penetrate the secondary emission sampling module placed between absorber materials. The generated secondary emission electrons are then multiplied in a similar way as the photoelectrons in photomultiplier tubes. Here, we report on the principles of secondary emission calorimetry and the results from the beam tests performed at Fermilab Test Beam Facility as well as the Monte Carlo simulations of projected, large-scale secondary emission electromagnetic calorimeters.

**Keywords:** secondary electron emission; radiation hardness; forward calorimetry; electromagnetic calorimetry





## 1. Introduction

The development of radiation-hard calorimeter systems is a long-standing problem. Despite the continuous need for this development, the amount of effort dedicated to R&D in this area is quite limited. In addition to a lack of novel developments, the currently operational detector systems suffer considerably from the lack of solid predictions of the effect of radiation on the active elements and the readout systems. Along this line, we attempted developing a novel, intrinsically radiation-hard calorimeter system based on the secondary emission (SE) principle. The detector modules envisaged will primarily utilize metal channel dynode chains, similar to that of the photomultiplier tubes, each coated with high secondary electron emission yield materials. The considered detector modules will be planar, of high granularity and tileable. The secondary emission technology is envisaged to be an asset for future implementations requiring radiation-hard, robust and cost-effective electromagnetic calorimeters [1,2].

Here we report on the principles of secondary emission calorimetry and the results from the beam tests of a dedicated secondary emission module constructed with basic principles. The Monte Carlo simulations of projected, large-scale secondary emission electromagnetic calorimeters are also presented.

## 2. Secondary Emission Detector Modules

In an SE detector module, SE electrons (SEe) are generated from an SE surface in the form of the cathode and the dynodes when charged hadronic or electromagnetic particles (shower particles) penetrate an SE sampling module either placed between absorber materials (Fe, Cu, Pb, W, etc.) in calorimeters or as a homogeneous calorimeter consisting entirely of dynode sheets as the absorbers. An SE cathode is a thin film, similar to the dynodes of photomultiplier tubes (PMTs). These films are typically simple metal oxides $Al_2O_3$, $MgO$, $CuO/BeO$, or other higher-yield materials. These materials are known to be very radiation-hard, as they are used in PMTs for up to 50 Grad dose and in accelerator beam monitors exposed to fluxes of higher than $10^{20}$ mip/cm$^2$ (see, e.g., [3] or [4]).

On the inner surface of a metal plate in vacuum, which serves as the entrance window to a compact vacuum vessel which is either metal or metal-ceramic, an SE film cathode is analogous to a photocathode, and the shower particles are similar to incident photons. The SEe produced from the top SE surface by the passage of shower particles, as well as the SEe produced from the passage of the shower particles through the dynodes, are similar to photoelectrons. The SEe are then amplified by sheets of dynodes, which could be metal meshes or other planar dynode structures. The SEe yield is a strong function of momentum, following dE/dx as in the Sternglass formula [5]. This variation with particle energy gives rise to quasi-compensation effects as the low-energy nuclear fragments of hadron showers have high yields, e.g., a 1 MeV alpha particle produces around 20 SEe. The comparison between SEe and photoelectrons should be emphasized: both are the result of dynode amplification. In a scintillation calorimeter, many photons are made per GeV, but typically only around 1–0.1% are collected and converted to photoelectrons; in an SE calorimeter, relatively few SEe from the shower particles are generated as the showers pass through the dynodes, but essentially all those SEe are amplified by the downstream dynodes. The result is that the statistics of photoelectrons and SEe are similar [6].

The construction requirements for an SE sensor module compared to the requirements for the construction of PMTs have several simplifications:

- The entire final assembly can be done in air. Dynodes used as particle detectors in mass spectrometers or in beam monitors cycle to air repeatedly.
- There are no critically controlled thin film vacuum depositions as in the case of photocathodes.
- Bake-out can be at refractory temperatures, unlike a photocathode, which degrades at temperatures higher than 300 °C.
- The SE module is sealed by normal vacuum techniques, and the necessary vacuum is 100 times worse compared to the PMTs.

The modules envisaged are compact, high gain, high speed, exceptionally radiation damage resistant, rugged, and cost effective, and can be fabricated in arbitrary tileable shapes.

## 3. Tests of the First SE Prototype Module

Due to the intrinsic similarities between the PMTs and the envisaged SE modules, the concept of an SE module can be validated by implementing relevant modifications to the PMTs. Since the photocathode functionality is not present in an SE module, the PMTs selected to construct the first SE module had excessive usage, and therefore had potentially degraded photocathodes. In addition, the photocathodes of the PMTs had the option of being disconnected from the multiplication chain so that the PMTs would not be responsive to any photons entering through or created at the window. Therefore, the entire dynode chain is utilized as SE surfaces. The largest signal is produced by an SEe produced at the first dynode (or the cathode).

The first SE prototype module was constructed with seven Hamamatsu single anode R7761 PMTs and was extensively tested at the Fermilab Test Beam Facility [7] with 4, 8 and 16 GeV electron beams. The characterization of the PMTs for the first SE sensor can be found in [8].

Figure 1 shows pictures of the first SE module. The module is designed to house the seven SE detectors in a closest-packed structure. A special electronics board was designed and produced for the first SE module. Figure 2 shows the circuit diagram of the electronics board for powering and readout of a single PMT.

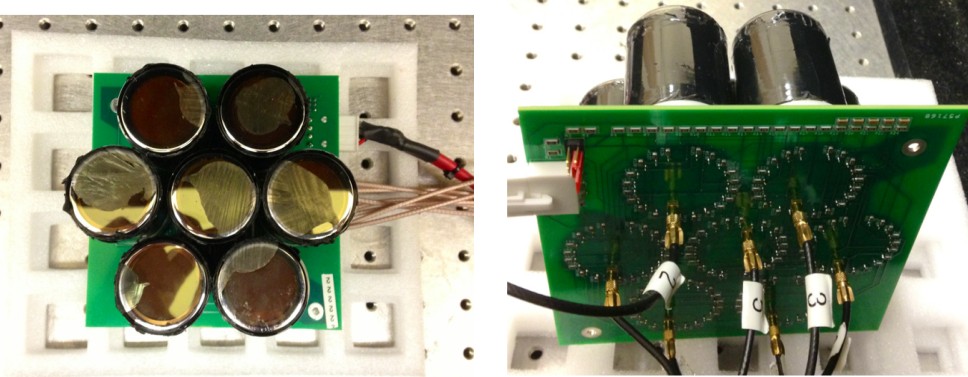

**Figure 1.** Pictures of the first SE module. Each sensor was 39 mm in diameter with an active window diameter of 27 mm. The length of the sensors was 50 mm.

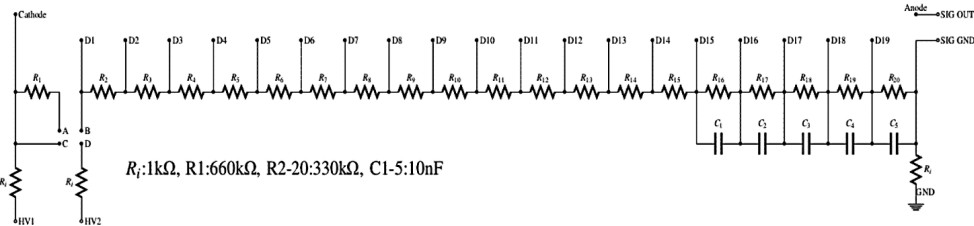

**Figure 2.** The circuit diagram of the baseboards for the powering and readout of a single PMT in the SE module.

Three different modes of operation exist on the baseboard for R7761 PMTs:

- Mode 1—normal divider mode: In this mode, the photomultiplier voltage divider chain is not modified and has equal potential differences across the dynodes, except the one across the cathode-first dynode (C–D1) gap, which is twice as large. This is the reference design from Hamamatsu.
- Mode 2—cathode-first dynode shorted: In this mode, jumpers on the board enable the bridging of R1, so that there is zero potential across the C–D1 gap (VC − VD1 = 0 V).
- Mode 3—cathode float mode: The design of the board allows the cathode to be separated from the remainder of the divider chain and be powered separately by another high voltage source. The potential across the C–D1 gap can also be adjusted such that it becomes positive with respect to the gap of D1–D2. If a second high voltage source is not used, the photocathode can still be charging up slightly. Dedicated tests resulted in no noticeable change in the response in particle beams when the photocathode was slightly reverse biased.

All of these modes can be examined in Figure 2, where the A-B bridge forms normal operation mode (Mode 1) with HV input on HV1, the B-C bridge forms Mode 2 with HV input on HV1, and the B-D bridge forms Mode 3 with HV input on HV2. Mode 2 was the default mode of operation for the beam tests.

Steel and tungsten absorbers were placed upstream of the SE module at increasing thicknesses to measure the shower development. With the 20 cm × 20 cm × 1.9 cm steel absorbers, all seven SE detectors were read out, and with the 3 cm × 3 cm × 0.35 cm tungsten absorbers, only the center module was read out. The lateral coverage of the SE module was not sufficient to produce a shower signal that scales with the shower depth with the steels absorbers. Therefore, the tests with the tungsten absorbers were taken as the baseline.

Figure 3 shows the response of the module to 8 (left) and 16 GeV (right) positrons with the tungsten absorbers. With a careful design of the trigger counters and the event selection based on the wire chambers, the electromagnetic shower profiles are accurately produced. The measurements (black) are also validated with Monte Carlo (MC) simulations (red). Figure 3 validates the concept of secondary emission sensors utilizing dynode chains similar to that of the photomultiplier tubes.

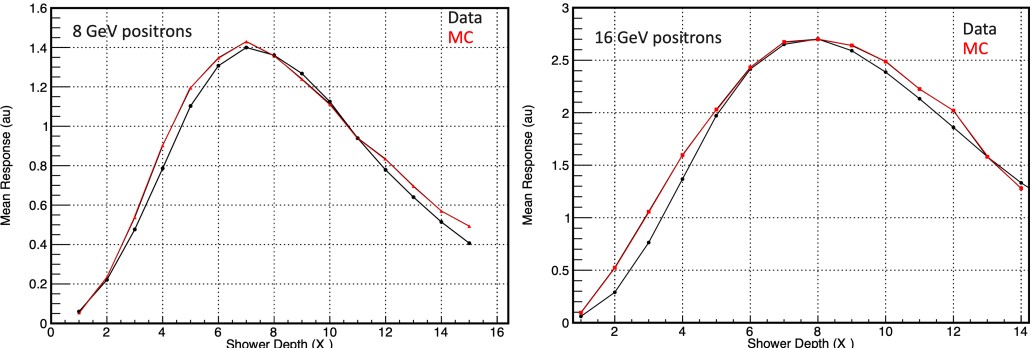

**Figure 3.** The response of the SE module to 8 (**left**) and 16 GeV (**right**) positrons with the tungsten absorbers.

## 4. Enhancement of Secondary Electron Emission

In order to enhance the production of secondary electrons in the SE modules, the cathode and the dynodes of the SE sensors can be made by coating the mesh copper foils with secondary emitters such as $Al_2O_3$, $SnO_2$, $TiO_2$ or $ZrO_2$. The coating can be done with vapor deposition techniques such as magnetron sputtering, for which $Al_2O_3$ and $TiO_2$ are very common targets.

Figure 4 (left) shows the simulated efficiencies for different thicknesses of $Al_2O_3$, $SnO_2$, $TiO_2$ and $ZrO_2$. The best performance is with a 100 nm thick $Al_2O_3$. The secondary electron emission efficiency is between 2 and 5% for all simulated secondary emitter coatings.

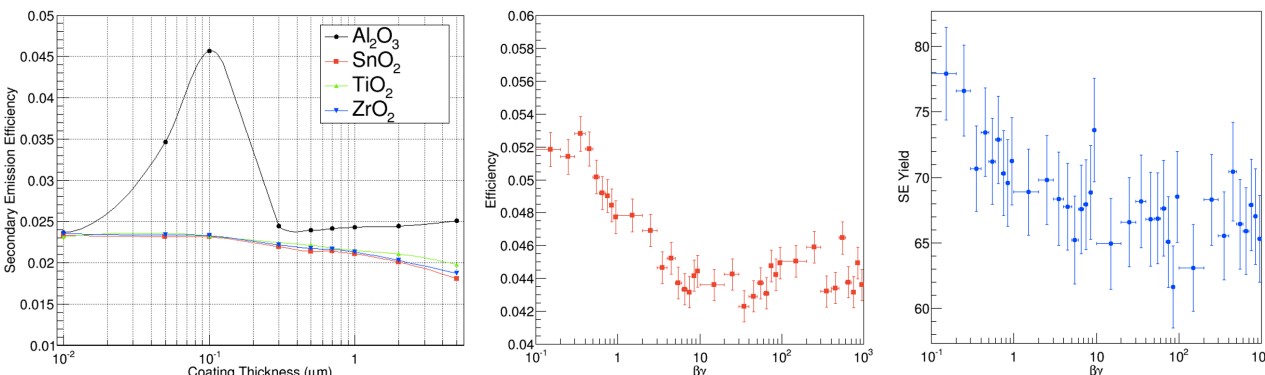

**Figure 4.** The simulated efficiencies for different thicknesses of $Al_2O_3$, $SnO_2$, $TiO_2$ and $ZrO_2$ (**left**), the secondary electron emission efficiency (**center**) and the SE yield (**right**) of the cathode once efficient as a function of the $\beta\gamma$ of the traversing particle for a 100-nm $Al_2O_3$-coated copper foil.

Figure 4 shows the secondary electron emission efficiency (center) and the secondary emission yield (right) of the cathode once efficient, i.e., there is at least one secondary electron produced at the cathode, as a function of the $\beta\gamma$ of the traversing particle for a 100-nm $Al_2O_3$-coated copper foil. The minimum ionization occurs around a $\beta\gamma$, of 40 which corresponds roughly to 4 GeV of muon energy. The average secondary electron yield is roughly around 68 with an increasing trend for lower $\beta\gamma$.

## 5. Projection of a Large-Scale SE Calorimeter Performance

In order to perform a simulation study for a large-scale SE calorimeter system, SE modules with 9-stage dynode chains were modeled. The number of dynodes was chosen so that the total number in one layer is minimum and the signal is still measurable. The dynodes are 150 μm apart and have 10–100 μm diameter holes, which are 50–100 μm apart. Figure 5 (left) shows the simulated charge spectrum for a 9-stage secondary emission device for a minimum ionizing particle that is efficient at the cathode. With an average of 300 fC, the signal can be recorded with commercial oscilloscopes.

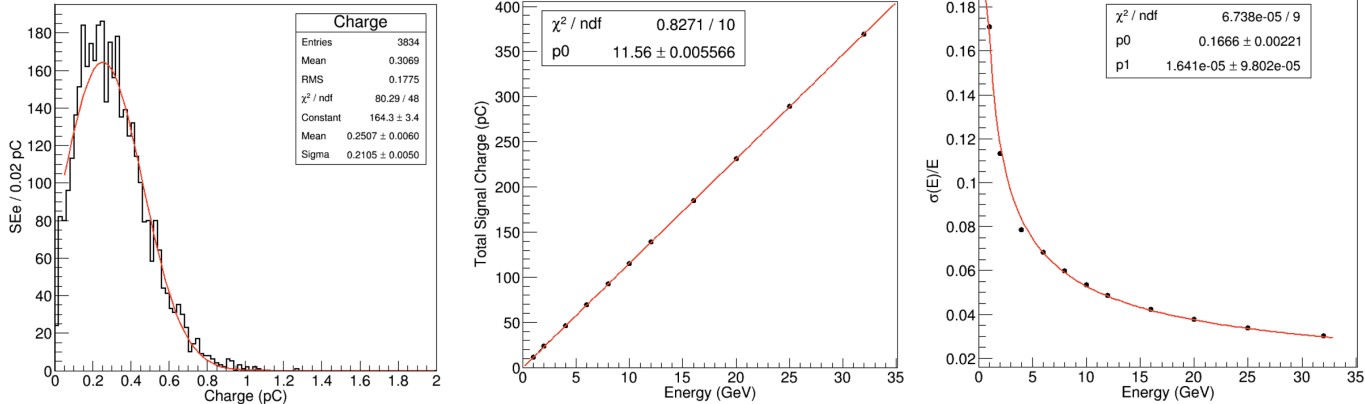

**Figure 5.** The simulated charge spectrum of a single layer (**left**), the MC predictions of the response linearity (**center**) and the energy resolution (**right**) of the 16-layer SE calorimeter prototype.

The electromagnetic response of an SE calorimeter prototype with 16 active layers interleaved with 1 $X_0$ tungsten absorbers was also simulated. The lateral size of the dynodes and the calorimeter layer was 1 m with no dead areas. The simulated electrons were normally incident on the front face of the calorimeter stack. Figure 5 (center and right) shows the MC predictions of the performance of the SE calorimeter prototype. The predictions are obtained for available Fermilab test beam energies of positrons/electrons for practical reference. The detector response is linear in the energy range of 1–32 GeV (center), and the electromagnetic energy resolution is obtained as $(16.7\%)/\sqrt{E}$ with a negligible constant term (right).

## 6. Conclusions

Secondary emission calorimetry is a feasible option particularly for electromagnetic calorimetry in high-radiation environments, as well as other implementations such as beam loss monitors and Compton polarimeters. The structure of the secondary emission sensors is quite similar to the dynode chain of photomultiplier tubes. The construction of the sensor modules have less strict vacuum requirements compared to photomultiplier tubes.

The first secondary emission sensor module was constructed with photomultiplier tubes with deactivated photocathodes. The preliminary tests validate the idea and suggest a full-scale secondary emission calorimeter prototype. The Monte Carlo simulations predict good response linearity and an energy resolution of $(16.7\%)/\sqrt{E}$ for a 16 layer calorimeter prototype up to 32 GeV. The secondary electron emission can also be enhanced by special surface coatings, such as $Al_2O_3$, applied on the dynodes.

Highly segmented readout for imaging calorimetry is possible with the envisaged secondary emission modules.

**Author Contributions:** Project administration, B.B., Y.O. and D.R.W.; Investigation, K.D., H.O., D.S., E.T. and J.W. All authors have read and agreed to the published version of the manuscript.

**Funding:** This research received no external funding.

**Data Availability Statement:** The data presented in this study are available on request from the corresponding author.

**Acknowledgments:** B.B. acknowledges support under Tübitak grant no. 118C224.

**Conflicts of Interest:** The authors declare no conflict of interest.

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
