# Peer review of "Secondary Emission Calorimetry"

_instruments, doi:10.3390/instruments6040048_

Round 1

Reviewer 1 Report

The authors present beam-test results and simulation studies from a secondary emission (SE) prototype module. SE calorimetry is promising for radiation-hard calorimeters. The manuscript is well-written and the experimental setup, measurement procedure and results are sufficiently clearly presented despite being very brief, presumably due to manuscript extension constraints. It is worth publishing in Instruments after addressing the minor comments below:  

- If available, a reference on SE calorimetry could be given in the introduction.

- L26: OF high granularity

- L72: Module Prototype -> Prototype Module. Same applies to other occurrences.

Author Response

On behalf of the authors, I would like to thank the referee for the constructive comments and suggestions. Please find my replies below:

The authors present beam-test results and simulation studies from a secondary emission (SE) prototype module. SE calorimetry is promising for radiation-hard calorimeters. The manuscript is well-written and the experimental setup, measurement procedure and results are sufficiently clearly presented despite being very brief, presumably due to manuscript extension constraints. It is worth publishing in Instruments after addressing the minor comments below:  

- If available, a reference on SE calorimetry could be given in the introduction.

====== Done.

- L26: OF high granularity

====== Done.

- L72: Module Prototype -> Prototype Module. Same applies to other occurrences.

====== Done.

Reviewer 2 Report

This conference poster proceeding shows interesting new concept for calorimetry. The paper is easy to read and needs only few minor improvements.

Here are some remarks to improve the manuscript

1) The references should be revisited.

    Reference 1 is useless and should be removed.

    Discussion of previous use of PMT-like devices as charged particle detectors could be mentioned. For example, this work : "A new electron-multiplier-tube-based beam monitor for muon monitoring at the T2K experiment", Y. Ashida et.al., PTEP 2018 (2018) 10, 103H01

   It would be useful for the reader to add a reference to the "Sternglass formula" at line 50.

2) The discussion on section 5 lacks information on the lateral design of the active layers in the simulation. How many SE devices were put per layer ? Are their input window circular like on figure 1 or with other shape ? What was the lateral dimension of the simulated calorimeter ?  What is the incidence angle of electrons entering the calorimeter ? This information is missing and a short description should be added.

 3) It is hard to distinguish the two curves in the plots of figure 3. At minimum, the authors might use different color also for the connecting lines.

4) Finally, minor corrections are suggested :

  a)  In caption of figure 1, the size of the devices (diameter, length) could be mentioned

  b) The term "cathode-first dynode" should be moved from line 95 to line 93 where it first appears.

 c) Line 132 and in caption of figure 4 :  the parenthesis opening should be moved between "once efficient" and "right".

Author Response

On behalf of the authors, I would like to thank the referee for the constructive comments and suggestions. Please find my replies below:

This conference poster proceeding shows interesting new concept for calorimetry. The paper is easy to read and needs only few minor improvements.
Here are some remarks to improve the manuscript
1) The references should be revisited.
    Reference 1 is useless and should be removed.
    Discussion of previous use of PMT-like devices as charged particle detectors could be mentioned. For example, this work : "A new electron-multiplier-tube-based beam monitor for muon monitoring at the T2K experiment", Y. Ashida et.al., PTEP 2018 (2018) 10, 103H01
   It would be useful for the reader to add a reference to the "Sternglass formula" at line 50.
===== Done.
2) The discussion on section 5 lacks information on the lateral design of the active layers in the simulation. How many SE devices were put per layer ? Are their input window circular like on figure 1 or with other shape ? What was the lateral dimension of the simulated calorimeter ?  What is the incidence angle of electrons entering the calorimeter ? This information is missing and a short description should be added.
===== Information was added.
 3) It is hard to distinguish the two curves in the plots of figure 3. At minimum, the authors might use different color also for the connecting lines.
===== Done.
4) Finally, minor corrections are suggested :
  a)  In caption of figure 1, the size of the devices (diameter, length) could be mentioned
  b) The term "cathode-first dynode" should be moved from line 95 to line 93 where it first appears.
 c) Line 132 and in caption of figure 4 :  the parenthesis opening should be moved between "once efficient" and "right".
====== All done.

Reviewer 3 Report

The manuscript is well written, the main strengths of this manuscript are that it addresses an interesting and timely question regarding the cost-efficient but robust calorimetry for the current and future collider experiments. It highlights several advantages of the novel techniques (using secondary emission principle as calorimeter) compared to the traditional method and then demonstrates the potential of using this novel technique as an alternative in high radiation environment based on a prototype design undergoing different electron beam energy tests and simulation. 

Title and abstract: The title is appropriate for the content of the manuscript. The abstract is concise and summarizes the key information of the manuscript although it would be better if the authors specify the location where the beam tests were performed in the abstract.

Body of the manuscript: The secondary emission calorimetry technique presentation is comprehensive but I feel that given the brevity of proceedings, the text would benefit more from additional references to support some of the statements and help to give background information to readers who are not too familiar with this new technique.  There are some minor points that should be clarified as well, as follows:

  1. Line 23: Please add a reference for the Secondary Emission principle
  2. Line 41-42: Please add a reference to support this statement "as they are used in PMTs for up to 50 Grad dose and in accelerator beam monitors exposed to fluxes of higher than 10^20 mip/cm^2".
  3. Line 50: Please add a reference for the Sternglass formula.
  4. Please clarify the last bullet point of the construction requirements for an SE Sensor Module, i.e. it was mentioned that the necessary vacuum for SE Sensor Module is 100 times higher than the PMTs. My understanding from reading the text is that the SE sensor module construction is simpler than PMTs, but if the construction requires a higher vacuum, wouldn’t that put more complications to the SE sensor module construction compared to the PMTs?
  5. Figure 4 right and line 132: please clarify what is meant by “once efficient"
  6. Line 134: “The average secondary electron yield is around 68 with an increasing trend for lower βγ”. Is the average calculated or just assumed to be 68 by looking at the data points trend in Figure 4c?
  7. Section 5: The simulation study is done by modeling 9-stage dynodes. Is there a particular reason why the projected large-scale simulation is not done following the SE sensor module prototype in data, which I believe is comprised of 19-stage dynodes? Also how many SE modules are modeled in the first simulation study?
  8. Figure 5 center and right: Do these figures demonstrate the results from the first simulation study or the “SE calorimeter prototype with 16 active layers interleaved with 1 X_0 tungsten absorbers”? Please label the type of simulation study these results are derived from in the caption if it's different.

Author Response

On behalf of the authors, I would like to thank the referee for the constructive comments and suggestions. Please find my replies below:

The manuscript is well written, the main strengths of this manuscript are that it addresses an interesting and timely question regarding the cost-efficient but robust calorimetry for the current and future collider experiments. It highlights several advantages of the novel techniques (using secondary emission principle as calorimeter) compared to the traditional method and then demonstrates the potential of using this novel technique as an alternative in high radiation environment based on a prototype design undergoing different electron beam energy tests and simulation. 

Title and abstract: The title is appropriate for the content of the manuscript. The abstract is concise and summarizes the key information of the manuscript although it would be better if the authors specify the location where the beam tests were performed in the abstract.

==== Done.

Body of the manuscript: The secondary emission calorimetry technique presentation is comprehensive but I feel that given the brevity of proceedings, the text would benefit more from additional references to support some of the statements and help to give background information to readers who are not too familiar with this new technique.  There are some minor points that should be clarified as well, as follows:

  1. Line 23: Please add a reference for the Secondary Emission principle
  2. Line 41-42: Please add a reference to support this statement "as they are used in PMTs for up to 50 Grad dose and in accelerator beam monitors exposed to fluxes of higher than 10^20 mip/cm^2".
  3. Line 50: Please add a reference for the Sternglass formula.
  4. Please clarify the last bullet point of the construction requirements for an SE Sensor Module, i.e. it was mentioned that the necessary vacuum for SE Sensor Module is 100 times higher than the PMTs. My understanding from reading the text is that the SE sensor module construction is simpler than PMTs, but if the construction requires a higher vacuum, wouldn’t that put more complications to the SE sensor module construction compared to the PMTs?
  5. Figure 4 right and line 132: please clarify what is meant by “once efficient"
  6. Line 134: “The average secondary electron yield is around 68 with an increasing trend for lower βγ”. Is the average calculated or just assumed to be 68 by looking at the data points trend in Figure 4c?
  7. Section 5: The simulation study is done by modeling 9-stage dynodes. Is there a particular reason why the projected large-scale simulation is not done following the SE sensor module prototype in data, which I believe is comprised of 19-stage dynodes? Also how many SE modules are modeled in the first simulation study?
  8. Figure 5 center and right: Do these figures demonstrate the results from the first simulation study or the “SE calorimeter prototype with 16 active layers interleaved with 1 X_0 tungsten absorbers”? Please label the type of simulation study these results are derived from in the caption if it's different.

==== All done.